# Betaine Reduces Lipid Anabolism and Promotes Lipid Transport in Mice Fed a High-Fat Diet by Influencing Intestinal Protein Expression

**DOI:** 10.3390/foods11162421

**Published:** 2022-08-12

**Authors:** Haitao Hu, Lun Tan, Xiaojiao Li, Jingjing Li, Caiyun Fan, Feng Huang, Zhao Zhuo, Kun Hou, Yinying Xu, Qingfeng Wang, Yongxin Yang, Jianbo Cheng

**Affiliations:** 1College of Animal Science and Technology, Anhui Agricultural University, Hefei 230036, China; 2College of Food Science and Engineering, Qingdao Agricultural University, Qingdao 266109, China

**Keywords:** betaine, intestinal, high-fat diet, lipid metabolism, proteomics

## Abstract

Betaine is more efficient than choline and methionine methyl donors, as it can increase nitrogen storage, promote fat mobilisation and fatty acid oxidation and change body fat content and distribution. Lipid is absorbed primarily in the small intestine after consumption, which is also the basis of lipid metabolism. This study was conducted to establish a mouse model of obesity in Kunming mice of the same age and similar body weight, and to assess the effect of betaine on the intestinal protein expression profile of mice using a proteomic approach. Analysis showed that betaine supplementation reversed the reduction in expression of proteins related to lipid metabolism and transport in the intestine of mice induced by a high-fat diet (HFD). For example, the addition of betaine resulted in a significant upregulation of microsomal triglyceride transfer protein (Mttp), apolipoprotein A-IV (Apoa4), fatty-acid-binding protein 1 (Fabp1) and fatty-acid-binding protein 2 (Fabp2) expression compared to the HFD group (*p* < 0.05), which exhibited accelerated lipid absorption and then translocation from the intestine into the body’s circulation, in addition to a significant increase in Acetyl-CoA acyltransferase (Acaa1a) protein expression, hastening lipid metabolism in the intestine (*p* < 0.05). Simultaneously, a significant reduction in protein expression of alpha-enolase 1 (Eno1) as the key enzyme for gluconeogenesis in mice in the betaine-supplemented group resulted in a reduction in lipid synthesis in the intestine (*p* < 0.05). These findings provide useful information for understanding the changes in the protein profile of the small intestine in response to betaine supplementation and the potential physiological regulation of diets’ nutrient absorption.

## 1. Introduction

Betaine, also known as trimethylglycine, is mainly found in crops grown under saline- and drought-tolerant conditions [1]; it is also a naturally occurring compound found in minimal quantities in some plants, animals and microorganisms [2,3]. Betaine was first discovered in the process of making molasses [4]. Derived from dietary intake on the one hand and from the oxidation of choline on the other, betaine is an important methyl donor in vivo and plays a key role in the maintenance of osmotic pressure homeostasis and normal fat metabolism in animals [5]. The addition of betaine to the diet of obese rats promoted the transfer of homocysteine to methionine in the rat liver via increasing the activity of key enzymes in the methionine metabolic pathway, such as cysteine methyltransferase. Therefore, concentrations of homocysteine in the liver of obese rats were reduced, and prevention of fat accumulation in the liver was achieved [6,7]. Betaine increased activity of antioxidant enzymes in the liver of ethanol-induced rats as well as alleviated plasma hyperhomocysteinemia in rats [8]. It was reported that a minimum of 4 g of betaine added to the diet each week reduced plasma levels of homocysteine in humans. [9]. In recent years, the role of betaine as a feed additive to modulate the digestion and absorption of nutrients and influence the body’s lipid metabolism in animals has been confirmed. Addition of 0.4% betaine to the diet of allogynogenetic gibel carp significantly improved growth performance and reduced lipid deposition, possibly by inhibiting the expression of obesity-associated genes [10]. The addition of 0.25% betaine to the diet improved pork quality and fat metabolism in Huanjiang pigs and was superior to equimolar amounts of glycine, and the authors concluded that betaine could partially replace glycine in fattening pig diets [11]. The latest study found that rumen-protected betaine (RPB) enhances the absorption of betaine in the intestine of ruminants, and betaine alters the distribution of fat [12]. Meanwhile, another study showed that lamb abdominal fat content decreased on a dose-dependent basis with dietary RPB, and that fat deposition in the longest dorsal muscle of lambs may be associated with regulation of the mTOR signalling pathway [13]. Feeding pigs with dietary betaine diets reduces carcass fat deposition and fat percentage [14]. Studies have shown that feeding mice with a diet rich in BET/choline can significantly reduce the energy and fat of apparent digestion [15].

Orally ingested free betaine was absorbed rapidly from the intestines, accumulated in the liver and kidney cells and then participated in body metabolism [16,17]. However, a careful and comprehensive analysis of the changes in intestinal proteins through betaine supplementation in mice has yet to be performed. The proteomics approach is a well-known powerful tool for investigating protein components at the entire protein level. In several studies, betaine supplementation in rat myocardial and adipose tissues and the main changes in the liver protein white matter level were investigated using a proteomics method [7,18]. Betaine was found to have a positive effect on the kidney functional protein expression of hyperuricemia mouse [19]. As we know, the intestine functions as an essential site for the absorption of small-molecule nutrients in mammals, and it is crucial to investigate the alterations in fat metabolism in the intestine of mice on a high-fat diet following betaine supplementation. Therefore, this experiment was conducted to investigate the effect of betaine on lipid metabolism in the intestine of mice on a high-fat diet by establishing a high-fat diet mouse model using nano-liquid chromatography combined with Q-Exactive proteomics, with the objective of understanding the effect of betaine on lipid metabolism from the perspective of the mouse intestine.

## 2. Materials and Methods

### 2.1. Animals and Study Design

A total of 36 male mice weighing 25.83 ± 2 g and 7 weeks old were randomly selected and assigned to 4 groups of 9 mice each. These groups of mice were fed a control diet (CD), a control diet with 1% betaine (CD + BET), a high-fat diet (HFD) and a high-fat diet with 1% betaine (HFD + BET), separately. Among these, 1% betaine in the CB and HFB groups was provided to the mice in the form of drinking water. The trial was continued for 5 weeks. The water was replaced every morning at 8 a.m. We ensured that all 4 groups of mice were provided food and water ad libitum. Details of the diet can be seen in the published articles by our research team [20].

### 2.2. Sample Collection and Preparation

When the experimental period was over, the mice were effectively anaesthetised and then dissected. Throughout the process, we were careful in ensuring each mouse suffered as little shock and pain as possible. Then, the small intestines (duodenum, jejunum and ileum) were separated, sliced and then placed in liquid nitrogen. Intestinal tissue of each mouse was repeatedly powdered in a ceramic textured mortar with liquid nitrogen. The powder was added to the lysis buffer (8 M urea, 2 M thiourea, 4% CHAPS, 20 mM Tris–base, 30 mM dithiothreitol), periodically vortexed in ice for 30 min and then centrifuged at 10,000× *g* for 15 min at 4 °C. The supernatant obtained in the previous step was mixed with 3-fold the volume of ice-cold acetone and then allowed to set at 4 °C for 30 min. After centrifugation, the protein globules were dissolved in 100 µL of 5 M urea and then blended with 4 times the volume of 40 mM NH_4_HCO_3_ solution. Concentrations of protein were determined using a Bicinchoninic Acid (BCA) assay. The protein samples were reduced with dithiothreitol solution (final concentration of 10 mM) for 1 h and then alkalised with iodoacetamide solution (final concentration of 50 mM) for 1 h in the dark. Subsequently, the protein samples were digested with sequencing-grade modified trypsin (protein: trypsin = 40:1) at 37 °C overnight. Tryptic peptides from each sample were collected.

### 2.3. Protein Identification

The tryptic peptides were analysed in this assay using EASY-nLC (Thermo Fisher Scientific, Waltham, MA, USA) and the Q-Exactive system (Thermo Fisher Scientific). Before the analytical separation, buffer A (0.1% acetic acid) was loaded onto a column named Aqua C18 (100 μm × 2 cm, 5.0 μm beads) at a flow rate of 5 μL/min. The peptide separations were performed on the analytical column over a gradient of 90 min. The separation procedure was as follows: 100% buffer A up to 8% buffer B (80% acetonitrile in buffer A) for 5 min, 8–20% buffer B for 55 min, 20–30% buffer B for 10 min, 30–100% buffer B for 5 min and 100% buffer B for 15 min.

The peptides eluted were then analysed using Q-Exactive (Thermo Fisher Scientific) in positive-ion and data-dependent mode. Mass spectrometry (MS) data were acquired with a full scan at a 400 m/z resolution power of 70,000 (precursor ion range of 300–1800 m/z). The first 20 MS/MS scans were conducted with a high energy dissociation (normalised collision energy: 27) at a resolution of 17,500, an isolation window of 2 m/z, the elimination setting of on isotopes and a dynamic exclusion time of 10 s.

### 2.4. Data Search

Primary data files were obtained with Xcalibur software (version 2.2, Thermo Fisher Scientific). Raw files were searched for mouse species in the UniProt database using PEAKS software (version 7.0, Bioinformatics Solutions Inc., Waterloo, ON, Canada). Parameters searched included trypsin specificity; aminomethylation of cysteine was designated as a fixed modification, and oxidation of methionine was designated as a variable modification. Simca-P software (Waters Corporation, Milford, MA, USA) was used for PCA analysis to detect differences in protein expression between the 4 groups. Cluster analysis of proteins differentially expressed in the intestine of the 4 groups of mice was performed using R programming language version 4.0 (University of Science and Technology of China, Hefei, China). GO annotation analysis of the genes that were differentially expressed was performed using DAVID version 6.7 (https://david.ncifcrf.gov/ (accessed on 14 September 2020)). The pathways involved in the 4 groups of differentially expressed proteins were enriched and visualised using the Kyoto Encyclopedia of Genes and Genomes (KEGG) (Kyoto University, Kyoto, Japan, https://www.kegg.jp/ (accessed on 14 September 2020)). The STRING database (version 10.5, https://string-db.org/ (accessed on 22 March 2021)) to build and detect possible protein–protein interaction (PPI) of differentially expressed proteins in the HFD + BET group was used.

Proteins identified in the intestines of the four groups of mice were quantified using label-free strategies with spectral intensities. Normalised spectral ion intensity for each sample in this experiment was subjected to one-way ANOVA using SPSS software (SPSS version 16.0, SPSS Inc., Chicago, IL, USA; IBM, Armonk, NY, USA). Proteins differentially expressed in the study were considered statistically significant with a *p*-value below 0.05 and at least a 2-fold change.

### 2.5. Western Blot Analysis

Proteins (25 g per sample) were detached by sodium dodecyl sulphate polyacrylamide gel electrophoresis (SDS PAGE) of 12%. Following electrophoresis until the protein bands were separated, the gels were transferred to polyvinylidene fluoride membranes and held for electrophoresis at a steady current (400 mA) for 60 min with a Mini Trans-Blot system (Bio-Rad, Hercules, CA, USA). After finishing the transfer, the gels were closed for 1 h at 25 °C in TBST (Tris-Hcl, NaC, tween20) containing 5% skimmed milk powder. The blocked membrane was incubated overnight with antiprimary antibody (1:1000 dilution, Abcam, Cambridge, UK, ab150113) at 4 °C. The membrane was washed three times with TBST and then incubated with horseradish-peroxidise-conjugated secondary antibody of Fabp1 (1:2000 dilution, Abcam, ab249184) and Acaa1a (1:3000 dilution, Abcam, ab110289) for 1 h at room temperature. Simultaneous testing of samples of all target proteins with the internal reference protein β-actin was carried out. Detection of antigen–antibody bound chromogenic reactions was achieved on a multifunctional gel imaging system with the Super Signal R West Pico Chemiluminescent Substrate. Western blot band results were quantified using Image J software (National Institutes of Health, Bethesda, MD, USA).

### 2.6. Statistical Analysis

The data were processed using Microsoft Office Excel software when comparing the differential protein expression in the intestines of the four groups of mice. The CD group was used as the control group for this experiment, and the protein expression in the intestine of the mice in this group was seen as a standard control, and the relative folds of protein expression in the intestine of the mice in the other three groups were calculated compared to the CD group. The relative folds of the differential proteins and the ratio of the grey scale values of the Western blot band were statistically analysed using one-way ANOVA and LSD multiple comparisons for each group of data using SPSS 16.0. *p* values less than 0.05 were considered to be significantly different. Finally, graphs were created using GraphPad Prism 8.0.

## 3. Results

### 3.1. Differences in Protein Expression among the Four Diets

The Nano-Liquid Chromatography–Mass Spectrometry (LC-MS) method was used to study the intestinal protein composition of mice. A total of 270 proteins were identified as remarkably differentially expressed in the CD, CD + BET, HFD and HFD + BET groups (fold change > 2, *p* < 0.05). Principal component analysis (PCA) of the differential protein levels in the mouse intestine showed that there were marked differences between the four groups (Figure 1A). PCA indicated that 73.77% of the variance in the dataset was explained by the first principal component, with an additional 14.61% explained by the second principal component. This indicated that HFD was the main factor leading to the separation and betaine was a secondary factor.

The heatmap analysis of the differential proteins is illustrated in Figure 1B. The protein profiles of the CD and CD + BET groups were more similar than those of the other two groups, while the protein components of the HFD and HFD + BET groups were obviously different. The result of the PCA was consistent with that of the heatmap analysis. For betaine, changes in the protein components that provided the largest contribution to the dietary treatment effects were mostly related to HF-induced mouse intestines.

### 3.2. Functional Analysis of Differentially Expressed Proteins

As shown in Figure 2, three sets of Gene Ontology (GO) were classified. The biological processes commonly associated with an HFD are the phosphorus metabolic process, catabolic process, oxidation–reduction process, carbohydrate derivative metabolic process and lipid metabolic process. With regard to cellular components, a high proportion of proteins were implicated in cellular processes and mitochondria. Molecular functional analysis revealed that 59.70% of the proteins identified were engaged in protein binding and 6.08% in lipid binding.

The KEGG pathway enrichment analysis revealed 14 significantly enriched path-ways (Figure 3). The preferential pathway for the largest number of significant differentially expressed proteins was the biosynthesis of antibiotics, followed by carbon metabolism, oxidative phosphorylation and protein processing in the endoplasmic reticulum. Most pathways can be related to antibiotics and energy metabolism in a variety of ways.

### 3.3. Protein–Protein Interaction Network Analysis

There were 98 proteins in the map (Figure 4), and the active interaction sources were set based on the experiments, databases and text mining. The interactive scores were ana-lysed at a confidence level of 0.4. In this protein network, alpha-enolase 1 (Eno1), methylenetetrahydrofolate dehydrogenase 1 (Mthfd1) and Rps15 presented more interactions than the other proteins that served as central hubs. These protein interactions are involved in lipid metabolism, fat transport and energy conversion. Coincidentally, protein expression of Eno1and Mthfd1 was significantly lower in the intestine of the HFD + BET group of mice.

### 3.4. Analysis of Differentially Expressed Proteins in the Intestines

As shown in Figure 5, several proteins involved in fat absorption and transport, such as microsomal triglyceride transfer protein (Mttp), apolipoprotein A-IV (Apoa4), fatty-acid-binding protein 1 (Fabp1) and fatty-acid-binding protein 2 (Fabp2), were found to increase in the intestines in the HFD + BET group compared with the HFD group (*p* < 0.05). Unlike in mice fed with the CD, Mttp protein was significantly increased in the CD + BET group (*p* < 0.05), while Apoa4, Fabp1 and Fabp2 were significantly reduced (*p* < 0.05). Acetyl-CoA acyltransferase (Acaa1a), which is associated with fatty acid metabolism and its relative abundance, was significantly increased in the HFD + BET group compared with the HFD group (*p* < 0.05). However, the expression level of Acaa1a decreased in the HFD group compared with the CD group (*p* < 0.05). The expression levels of Eno1 and Mthfd1 were significantly higher in the HFD group than the other studied groups (*p* < 0.05). Finally, we verified the protein expression of Fabp1 and Acaa1a using Western blot analysis, and the results show that the proteomic results are consistent with the quantitative results of Western blot; that is, the protein expression of Fabp1 and Acaa1a were significantly elevated in the intestine of mice in the HFD + BET group (*p* < 0.05) (Figure 6).

## 4. Discussion

This study aimed to investigate the effects of HFD intake on the protein components in the intestines of mice and explore the changes in nutrient absorption and metabolism caused by betaine supplementation. Once the lipids passed into the intestine, the lipids would be digested into small molecules by lipase enzymes, and the lipolytic products would then be absorbed by the intestinal epithelial cells. Eventually, celiac particles were formed in the intestinal cells and secreted into the circulating body fluid [21,22]. The assembly of these triglyceride-rich lipoproteins in intestinal cells, including the lipolytic products, is absorbed and transported from the bristles to the endoplasmic reticulum by synthesis of FABPs and triglycerides (TG) [23,24]. Lipids and Apoa4 are packaged as lipoprotein particles and transported into Golgi for secretion [25]. Mttp is a heterodimeric lipid transfer protein that facilitates the transport of four major lipids (cholesterol, phospholipids, triglycerides and cholesterol esters) to newly generated lipoprotein particles by assisting apolipoprotein B (ApoB) folding and translocation while preventing the newly generated ApoB from being degraded by proteasome [26]. Mttp is essential for cholesterol absorption. Intestinal Mttp ablation increases intestinal lipids and reduces plasma lipoproteins [27]. Previous experiments have shown that in mice fed with an HFD for a short period, most of the insulin-related mRNA levels, including Mttp and Apoa4, were reduced [28]. Although the HFD induced a downregulation of Mttp mRNA expression associated with very low density lipoprotein (VLDL) in mice, the expression was significantly increased after 2% betaine supplementation due to a decrease in the mean methylation level of the CpG site of Mttp by betaine [29]. We also found that Mttp was downregulated in the small intestines of mice fed with an HFD with betaine compared with mice fed with an HFD without betaine.

The primary function of Apoa4 is to bind to lipid molecules, and Apoa4 is expressed in the mammalian gut and in the liver of rodents [30]. Apoa4 in the intestine entered the circulation via the chylomicrons and interfered with the metabolism of the lipoproteins [31]. It was found that due to an increased expression of Apoa4, which in turn promoted the transport of celiac particles, and accelerated the secretion of triglycerides out of the liver, the symptoms of steatosis were reduced [32]. Although several studies have found that Apoa4 expression is elevated after HFD, the chronic intake of HFD can lead to different responses of intestinal Apoa4 to dietary lipid levels [33]. Feeding rats with an HFD can increase Apoa4 in plasma in a short time, but the expression level of Apoa4 decreased with the prolongation of time in the duodenum mucosa of HFD-induced mice [28]. The synthesis and level of Apoa4 in the jejunum mucosa also showed a time-dependent adverse effect on fat feeding [34]. Unlike in feeding with a low-fat diet, rodent overfeeding significantly reduced the gene expression of the Apoa4 in the jejunum and hypothalamus [35]. Furthermore, Apoa4 is an anorexic protein produced in the intestine and brain that has the effect of reducing feeding [31]. Studies have shown that natural Apoa4 administration inhibits food intake and that intraperitoneal administration of recombinant Apoa4 alone significantly reduced short-term food intake in rats [36]. Additionally, another study showed that Apoa4-induced dietary depression led to inhibition of the PI3K/Akt pathway in the ventromedial hypothalamus [37]. In our study, Apoa4 expression was significantly increased in the HFD + BET group of mice. It is possible that this accelerated the transport of fat in the intestine of mice in the HFD group and reduced the accumulation of lipids in the intestine of mice on a high-fat diet. Moreover, the elevation of Apoa4 may also have been a factor in the lower food intake of mice in the HFD + BET group [20]. However, this phenomenon needs further investigation.

The intestine mediates the entry of nutrient lipids, which are one of the major components of beta-oxidation [38,39]. Fabp2 (intestinal) has a single site that binds to the fatty acid (FA) ligand, which has a high affinity for binding saturated and unsaturated FAs, but it seems to have a somewhat lower affinity for binding unsaturated FAs than Fabp1 [40]. Fabp2 and Fabp1 can bind large numbers of FAs from the intestine to the cytoplasm in a fed state to prevent excessive accumulation of unbound FAs in the cytosol and to maintain the steady-state level of unbound FAs [41]. Fabp2 not only binds to FAs but also involves the use of intestinal fatty acids, the overexpression of which enhances the β oxidation of mitochondria in intestinal epithelial cells and reduces fat synthesis [42]. Studies have shown that weight gain and blood-free TG concentration increased in Fabp2 gene-deficient mice compared with wild-type mice [43]. The addition of betaine may increase the use of enteric fatty acids, leading to a decrease in appetite glucose-dependent insulinotropic polypeptide secretion, which reduces the food intake of mice and adipose tissue accumulation [44]. In our study, Fabp2 was increased in the HFT + BET-diet-fed mice. This may be due to the transfer of FAs to intestinal tissue and to further FA metabolism. A previous study found that Acaa1a appeared to be a core obesity gene, which is associated with fatty acid β oxidation in peroxisomes [45]. In mice fed an HFD rich in corn oil, the expression of Acaala in the intestinal tract was inhibited, and the ability of intestinal fatty acid beta-oxidation was inversely proportional to the content of corn oil [46]. Alcohol-induced fatty liver also reduced fat metabolism and the gene expression of the Acaa1a [47,48]. In our experiment, a significant increase in the protein expression of Acaa1a improved the oxidation of FAs in the HFT + BET-diet-fed mice.

Eno1 is a key enzyme in the process of glycolysis, catalysing the reversible reaction of 2-phosphoglycerate dehydration to form enol pyruvate. Compared with a low-concentration diet, the expression of Eno1 protein in normal goat liver was significantly upregulated in a high-concentration diet [49]. Eno1 is conducive to the synthesis of fat to provide precursor materials and increase fat accumulation. Compared with the control diet, the expression of the Eno1 protein in normal mouse kidney was significantly downregulated in a high methionine diet [50]. In our study, the expression level of Eno1was also significantly decreased in the HFT + BET-diet-fed mice, which could contribute to a decrease in the accumulation of fat.

Mthfd1 is the main source of thymidine synthesis of a carbon unit. Folic-acid-mediated carbon metabolism is a metabolic network of de novo synthesis of DNA bases and the remethylation of homocysteine and methionine [51]. Studies have shown that the Mthfd1 gene was necessary in mice and that it confirmed the basic role of mitochondria in the cytoplasm of 1C metabolism in the form of formate. Compared with that of Mthfd1gt/+ (Mthfd1 expression of 50%) mice, the weight gain of wild-type mice was significantly increased [52]. The results indicate that an increase in the expression of Mthfd1 was conducive to the accumulation of fat. In our study, the expression of Mthfd1 was highest in the HFD group, and the expression level of Mthfd1 was significantly decreased after betaine supplementation. Our results suggest that Eno1 and Mthfd1 may be involved in fat accumulation and that their expression levels could be induced by a diet with betaine supplementation.

## 5. Conclusions

In conclusion, the results indicate that supplementation of betaine to mice on a normal diet caused alterations in the expression of proteins related to intestinal glucolipid metabolism. The more significant results show that feeding mice with an HFD induced intestinal protein expression associated with fat transport and gluconeogenesis. Supplemental betaine partly reversed HFD-induced low lipid transport and increased lipid anabolism by regulating the protein expression levels of the intestinal tissue. These findings may provide clues that help us to understand the effect of dietary supplementation of betaine on lipid metabolism induced by an HFD and help clarify the role of intestinal protein expression in controlling systemic lipid metabolism.

## Figures and Tables

**Figure 1 foods-11-02421-f001:**
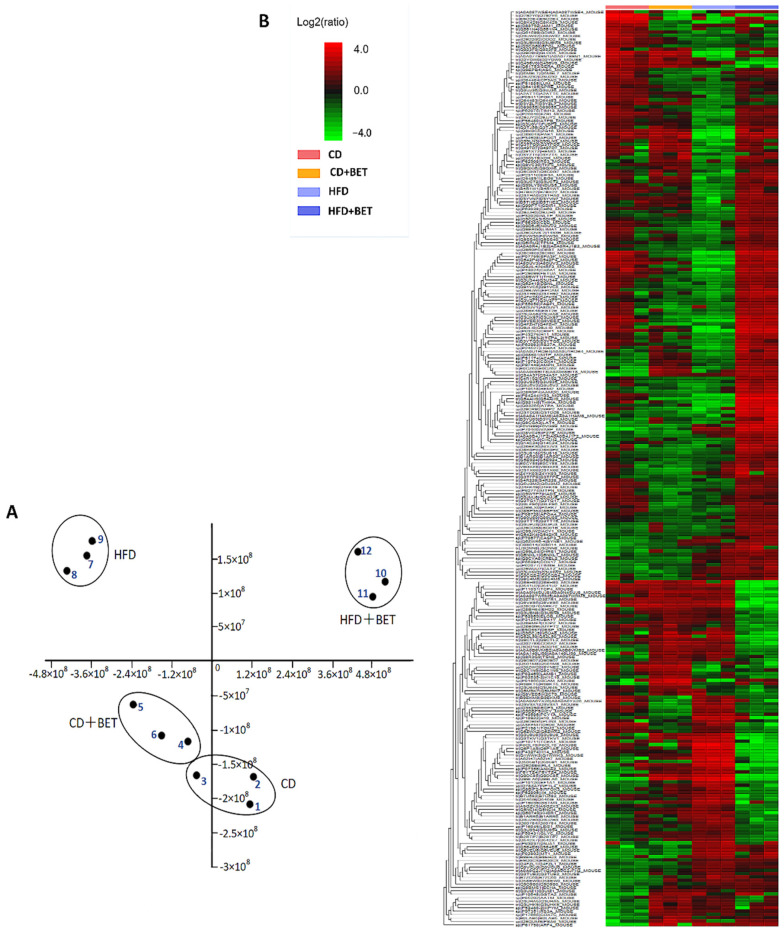
Results of principal component analysis (PCA) and cluster analysis of intestinal protein profiles in mice of CD, CD + BET, HFD and HFD + BET groups. (**A**) PCA revealed significant sepa-ration of intestinal proteomic features in the four groups of mice. The points 1, 2 and 3 in the dia-gram represent group CD; points 4, 5 and 6 represent group CD + BET; points 7, 8 and 9 represent HFD group and points 10, 11 and 12 represent group HFD + BET. (**B**) Heat map of differential protein profiles among 4 groups of mice intestines. The red indicates high concentration levels of proteins and the green indicates low concentrations of proteins.

**Figure 2 foods-11-02421-f002:**
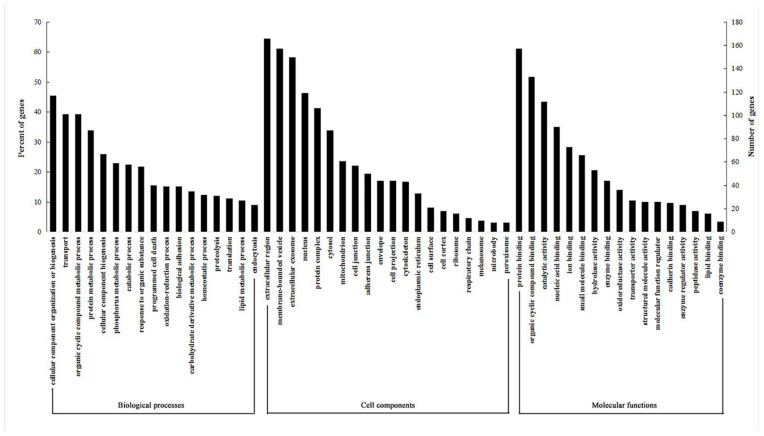
Functional annotations of GO for the 263 unigene products are classified into 3 major categories and 55 subcategories. The right *Y* axis indicates the number of unigenes products in a particular category. The left *Y* axis represents the percentage of genes in a particular category within each major category.

**Figure 3 foods-11-02421-f003:**
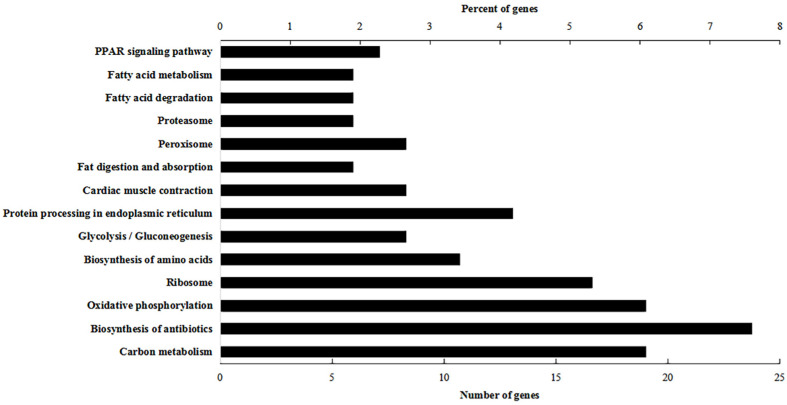
KEGG classification of the 263 protein functionalities and classified as 14 major pathways. The percentage of genes in a specific functional cluster is indicated above the *X* axis. The number of genes in a category is indicated below the *X* axis. *Y* axis represents differentially expressed genes products enriched in pathways.

**Figure 4 foods-11-02421-f004:**
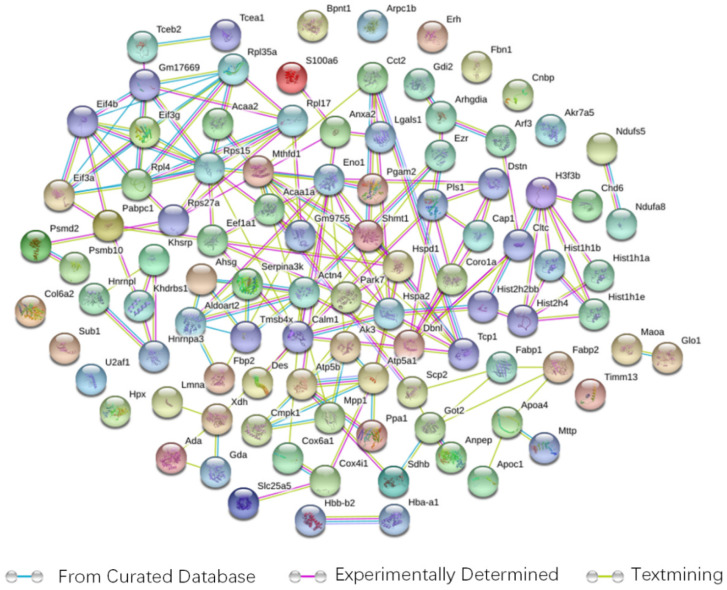
The protein interaction network of differential proteins. There were 98 points (proteins) and 164 interactions between these proteins. Network nodes represent proteins, while the edges between nodes express the association between proteins and the colour of the edges indicates the type of interactivity.

**Figure 5 foods-11-02421-f005:**
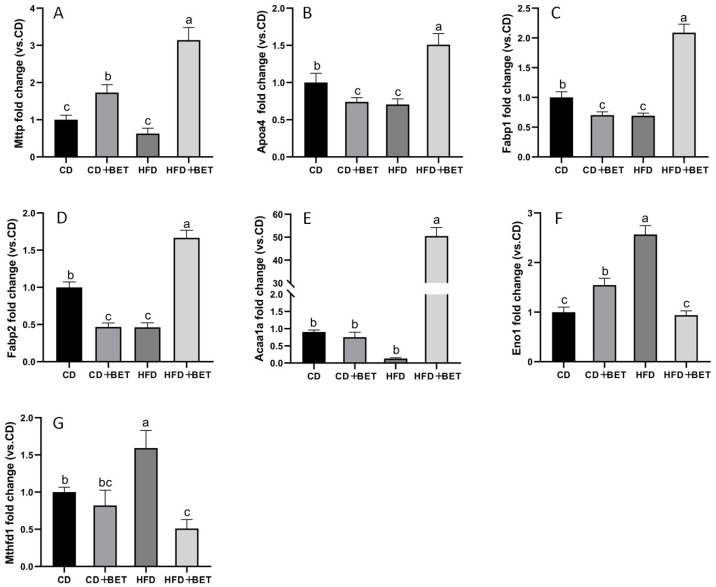
Effect of four diets on relative intestinal protein density of Mttp (**A**), Apoa4 (**B**), Fabp1 (**C**), Fabp2 (**D**), Acaa1a (**E**), Eno1 (**F**) and Mthfd1 (**G**). a–d: The lowercase letters above each bar reflect the difference between the groups. If the lowercase letters above the bars are different, it means that there is a significant difference between the two groups (*p* < 0.05), and if the same letters are present, it means that there is no significant difference between the two groups (*p* > 0.05).

**Figure 6 foods-11-02421-f006:**
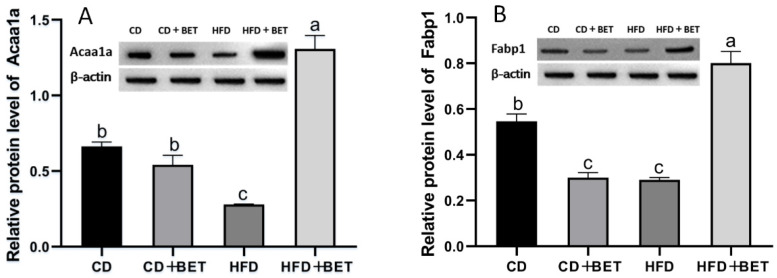
Verification of the expression of intestinal protein in the different groups. The Western blot bands of Acaa1a and Fabp1. β-actin was used as reference. a–d: The lowercase letters above each bar reflect the difference between the groups. If the low-ercase letters above the bars are different, it means that there is a significant difference between the two groups (*p* < 0.05), and if the same letters are present, it means that there is no significant difference between the two groups (*p* > 0.05).

## Data Availability

Data is contained within the article.

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
