# Peer review of "Betaine Reduces Lipid Anabolism and Promotes Lipid Transport in Mice Fed a High-Fat Diet by Influencing Intestinal Protein Expression"

_foods, 2022, doi:10.3390/foods11162421_

Round 1
Reviewer 1 Report
Title – on my opinion the title does not represent the results of the study. The authors have not measured lipid accumulation and/or metabolic disorders parameters. The manuscript represents proteomics data. That is why I would recommend change of the title.
Introduction – the aim should more clearly defined
Materials and methods – description of anesthesia should be described; Ethics permission should be provided; description of statistical analyses should be provided
48 – BET – the full name required
87 – Table 1
91 - the table caption
Figure 1 – the quality of the text to the heat map is poor and the text is not readable
Figure 2 - the quality of the text in figures is not satisfying, too
Figure 4 – Aatabase?
Figures 5 and 6 – I would recommend another form of presentation of statistical comparisons
255 – “This is due to the fact that BET significantly increased the average methylation level of the Mttp promoter” – how had been this proved; citation required
266 – “Feeding rats with HFD can increase Apoa4 in plasma in a short time, but the expression level of Apoa4 decreased with the prolongation of time” – in which organ/tissue?
Conclusions – Based on the results, I understand that BET has a beneficial effect in CD diet too, not only in HFD.
I would recommend some language improvement, e.g.:
43 – “in the vivo”; 47 – “ethanol induced rat liver”; 53 – “lipid genes”; 243 – “FABPs and triglycerides (TG) synthesis”; 253 – “mice induced by HFD”; 258 – “Apoa4 is a lipid-binding relative protein”; 275 – “hypothalamic”; 278 – “adipose”
Author Response
Dear Reviewer:
Thank you very much for your valuable comments and advice on our manuscripts. Our revised manuscript and the response to the section editors’ comments were enclosed. The manuscript title is “Betaine Attenuates High-Fat Diet-Induced Lipid Accumulation and Metabolic Disorders in Mice by Influencing Intestinal Protein Expression” (This is a title that was initially submitted and has now been corrected.) (foods -1825441). We have taken your comments into careful consideration. We consider all these comments to be very worthwhile and helpful for the improvement of our paper. Based on your suggestions I have revised the article and hope that our revisions will be satisfactory. Our revisions to the manuscript were marked up using the “Track Changes” function in uploaded revision manuscript. Please check the uploaded revised manuscript. Sincere thanks again to you!
General: Comments and Suggestions for Authors:Title – on my opinion the title does not represent the results of the study. The authors have not measured lipid accumulation and/or metabolic disorders parameters. The manuscript represents proteomics data. That is why I would recommend change of the title.
Response: We believe that your suggestion is correct and we have revised the title of the article. Now we understand that the title is actually not a very accurate and specific summary of the results of this experiment. Furthermore, it is both inaccurate and unspecific for us to interpret the reduction in expression of proteins related to intestinal lipid anabolism caused by betaine supplementation in mice on a high-fat diet as a disorder of lipid metabolism. Therefore, I amended our incorrect statement in 6. Conclusion. Please check the uploaded revised manuscript (Lines 374-375). Meanwhile, I have also revised the title to read: Betaine Decrease Lipid Anabolism and Increase Lipid Transport in Mice fed a High-Fat Diet by Influencing Intestinal Protein Expression. Please check the uploaded revised manuscript (Lines 2-4).
Point 1: Introduction – the aim should more clearly defined.
Response 1: I have clearly defined the aim of this experiment in the introduction. Specifically amended to read as follows: As we know, the intestine functions as an essential site for the absorption of small molecule nutrients in mammals, and it is crucial to investigate the alterations in fat metabolism in the intestine of mice on a high-fat diet following betaine supplementation. Therefore, this experiment was conducted to investigate the effect of betaine on lipid metabolism in the intestine of mice on a high-fat diet by establishing a high-fat diet mouse model using nano-liquid chromatography combined with Q-Exactive proteomics, with the objective of understanding the effect of betaine on lipid metabolism from the perspective of the mouse intestine. Please check the uploaded revised manuscript (Lines 76-83).
Point 2: Materials and methods – description of anesthesia should be described; Ethics permission should be provided; description of statistical analyses should be provided
Response 2: With respect to your three proposed modifications, I have responded to each of them as follows:
- – description of anesthesia should be described;
I have added specific steps to anesthetize mice in the 2.2. Sample collection and preparation section of the article. Please check the uploaded revised manuscript (Lines 104-107).
- – Ethics permission should be provided;
In accordance with the guidelines of the journal “foods”, a statement is required at the end of the paper in cases where the ethics and welfare of laboratory animals are involved in the experiment. The content of the ethical license can be found in the Institutional Review Board Statement following the conclusion of the article. Please check the uploaded revised manuscript (Lines 390-394).
- – description of statistical analyses should be provided
On the basis of your suggestion, we have added 2.6. Statistical analysis to the section 2. Materials and methods. The specific methods of data analysis have been written down. Please check the uploaded revised manuscript (Lines 176-186).
Point 3: 48 – BET – the full name required
Response 3: Thank you very much for your advice. With careful consideration, I have replaced all the “BET” in the text (except where the expression grouping is concerned) with “betaine” to avoid misunderstandings. Of course, the BET in line 48 that you mention has been replaced with betaine. Please check the uploaded revised manuscript (Line 51).
Point 4: 87 – Table 1
Response 4: Table has been revised as Table 1. Moreover, where this table is mentioned, it has been amended to Table 1. Please check the uploaded revised manuscript (Line 97 and line 101).
Point 5: 91 - the table caption
Response 5: This error has been revised. Please check the uploaded revised manuscript (Line 97 and line 101).
Point 6: Figure 1 – the quality of the text to the heat map is poor and the text is not readable.
Response 6: I have adjusted the font of the legend for Figure 1B to be a little larger, which will hopefully improve the readability of the image. Furthermore, I want to explain the analysis of the heat map as follows: the cross coordinates in Figure 1B represent the samples (that is, the four groups in this experiment, which can be distinguished by the colors at the top of Figure 1B). The results of the cluster analysis are shown in Figure 1B in the vertical direction. Please check the uploaded revised manuscript (Line 204).
Since we have analyzed protein profiles with differences in the four groups, the purpose is to represent the effect of high fat diet on normal diet as well as the betaine on intestinal protein expression in obese mice as a whole. Due to the large number of differential proteins, we can only clearly see the code of each protein when the picture is enlarged. However, we can clearly see the changes in the intestinal protein profiles of the four groups of mice by the color changes in the graphs in the Figure 1B, especially when comparing the first group with the third group and the third group with the fourth group. Owing to the above, I have inserted the original image of Figure 1B in the uploaded supplementary material. Please check “Figure 1B heatmap” in the supplementary material, where you can see the detailed differential protein numbers by enlarging the image. I would like to hope that my response to your question will be satisfactory to you. Please check the file "Figure 1B heatmap" in the uploaded zip file of the supplementary material.
Point 7: Figure 2 - the quality of the text in figures is not satisfying, too
Response 7: Figure 2 in this article shows the results by GO annotation of the differential proteins, mainly includes biological processes, cellular components and molecular function. Because there are more items enriched in these three directions, the images containing the content are only arranged in the horizontal direction, and when the indentation of the lines is adjusted and the image is enlarged a little, the specific items with the different proteins can be seen. However, I am not sure if this would be contrary to the requirements of the journal "foods" for the layout of the manuscript. Please check the uploaded revised manuscript (Line 217).
Point 8: Figure 4 – Aatabase?
Response 8: Many thanks for your careful review and we do apologize that this was a writing error. It should be written correctly as Database, not Aatabase, and I have made changes in Figure 4. Please check the uploaded revised manuscript (Line 241).
Point 9: Figures 5 and 6 – I would recommend another form of presentation of statistical comparisons
Response 9: We do appreciate your advice and we believe that your proposal is also quite guiding. We have also tried to present the expression of several differential proteins in specific numbers that we showed in the article. However, because the magnitude of the data is large and the statistics in tabular form are very voluminous, we finally chose to present the final statistics in graphical form. I wish to explain to you the purpose of making a graph like this: we use the CD group as the control group for the whole experiment and then set the intestinal protein expression in this group as a standard control and compare the other three groups with it, so that we can clearly observe the differences and changes in intestinal protein expression in the all groups. Therefore, we have not reconstructed Figures 5 and we have also added the specific methodology for our statistics in the section on 2. materials and methods. Please check the uploaded revised manuscript (Lines 176-186 and lines 265-269). Hopefully I have described the statistical methods of these two graphs clearly, thus making them more readable, and also hopefully to your satisfaction.
Point 10: 255 – “This is due to the fact that BET significantly increased the average methylation level of the Mttp promoter” – how had been this proved; citation required
Response 10: Thank you very much for alerting me to the errors that have occurred here. I sincerely apologize for my crude writing, having read reference 28 carefully again I found an error in my previous statement. In fact, the expression of Mttp mRNA is negatively correlated with the level of methylation. I have now revised the issue you mentioned. Please check the uploaded revised manuscript (Lines 291-294).
Point 11: 266 – “Feeding rats with HFD can increase Apoa4 in plasma in a short time, but the expression level of Apoa4 decreased with the prolongation of time” – in which organ/tissue?
Response 11: Upon review of the literature, this change of Apoa4 expression occurred in the duodenal mucosa of mice induced by a high fat diet. I have revised this issue in the revised version, please check the uploaded revised manuscript (Line 311).
Point 12: Conclusions – Based on the results, I understand that BET has a beneficial effect in CD diet too, not only in HFD.
Response 12: Many thanks for your suggestion. According to the results of the experiment, betaine supplementation for mice on a normal diet did have an effect on protein expression in the intestine of mice, but this article focused on the analysis of the effect of betaine on protein expression in the intestine of mice on a high fat diet. Hence I have added this finding to the conclusion section as you mentioned. Please check the uploaded revised manuscript (Lines 371-373).
Point 13: I would recommend some language improvement, e.g.: 43 – “in the vivo”; 47 – “ethanol induced rat liver”; 53 – “lipid genes”; 243 – “FABPs and triglycerides (TG) synthesis”; 253 – “mice induced by HFD”; 258 – “Apoa4 is a lipid-binding relative protein”; 275 – “hypothalamic”; 278 – “adipose”
Response 13: We are grateful to you for pointing out these humble but vital errors of expression, and we believe it will be critical to improve the quality of our manuscripts. The specific amendments are as follows:
- "in the vivo" has been revised as " in vivo", Please check the uploaded revised manuscript (Line 44);
- “ethanol induced rat liver” has been revised as “the liver of ethanol-induced rats”. Please check the uploaded revised manuscript (Line 49-50);
- “expression of lipids genes” has been revised as “expression of obesity-associated genes”. Please check the uploaded revised manuscript (Line 56);
- “FABPs and triglycerides (TG) synthesis” has been revised as “synthesis of FABPs and triglycerides (TG)”. Please check the uploaded revised manuscript (Line 282);
- The statement “mice induced by HFD” has been revised. The complete sentence is expressed as: Although HFD induced a down-regulation of Mttp mRNA expression associated with very low density lipoprotein (VLDL) in mice, the expression was significantly increased after 2% betaine supplementation due to a decrease in the mean methylation level of the CpG site of Mttp by betaine [28]. Please check the uploaded revised manuscript (Lines 291-294);
- The statement “Apoa4 is a lipid-binding relative protein” has been revised. The complete sentence is expressed as: The primary function of Apoa4 is to bind to lipid molecules and Apoa4 is expressed in the mammalian gut and in the liver of rodents. Please check the uploaded revised manuscript (Lines 301-302);
- “hypothalamic” has been revised as “hypothalamus”. Please check the uploaded revised manuscript (Line 320);
- “adipose” has been revised as “lipids”. Please check the uploaded revised manuscript (Line 323).

Reviewer 2 Report
The proposed manuscript entitled “Betaine Attenuates High-Fat Diet-Induced Lipid Accumulation 2 and Metabolic Disorders in Mice by Influencing Intestinal Protein Expression” suits very well the aim and scope of the journal Foods. The information gathered using in vivo set-ups is very valuable, and interesting.
There are a few issues I would like to address. I recommend to add an additional sub-chapter to the materials and methods section to describe in a more comprehensive manner the statistical analysis that were performed. These should also include to any statistical analysis that were performed using the western blot data.
According the journal submission guidelines, we suggest the authors include supplementary materials consisting in the uncropped western blots that were used as source for the Figure 6. Also complete information regarding the primary and secondary antibodies used (names and codes) should be provided.
There is no reference to Figure 5 in the text. In figures 5 and 6 the legends do not indicate what the letter above the histogram bars represent. In the legend of figure 5 the last statement indicates ‘Same as below’, however there is not any additional clarifying information.
Why were only Fabp1 and Acaa1a protein expression verified by western blot?
Author Response
Dear Reviewer:
Thank you very much for your valuable comments and advice on our manuscripts. Our revised manuscript and the response to the section editors’ comments were enclosed. The manuscript title is “Betaine Attenuates High-Fat Diet-Induced Lipid Accumulation and Metabolic Disorders in Mice by Influencing Intestinal Protein Expression” (foods-1825441). We have taken your comments into careful consideration. We consider all these comments to be very worthwhile and helpful for the improvement of our paper. Based on your suggestions I have revised the article and hope that our revisions will be satisfactory. Our revisions to the manuscript were marked up using the “Track Changes” function in uploaded revision manuscript. Please check the uploaded revised manuscript. Sincere thanks again to you!
General: The proposed manuscript entitled “Betaine Attenuates High-Fat Diet-Induced Lipid Accumulation 2 and Metabolic Disorders in Mice by Influencing Intestinal Protein Expression” suits very well the aim and scope of the journal Foods. The information gathered using in vivo set-ups is very valuable, and interesting.
Point 1: There are a few issues I would like to address. I recommend to add an additional sub-chapter to the materials and methods section to describe in a more comprehensive manner the statistical analysis that were performed. These should also include to any statistical analysis that were performed using the western blot data.
Response 1: Thank you very much for your suggestion. In combination with your and another reviewer's comments, I have added appropriate operational details to the materials and methods section of the article in the sections 2.2. Sample collection and preparation, 2.4. Data search and 2.5. Western blot analysis. Please check the uploaded revised manuscript (Lines 104-107, lines 146-155 and lines 174-175). In addition, I have completed the materials and methods section with 2.6. Statistical analysis to make the article more readable. Please check the uploaded revised manuscript (Lines 176-186).
Point 2: According the journal submission guidelines, we suggest the authors include supplementary materials consisting in the uncropped western blots that were used as source for the Figure 6. Also complete information regarding the primary and secondary antibodies used (names and codes) should be provided.
Response 2: The brands and dilution ratios of the three antibodies used in this test are described in 2.5 Western blot analysis.The brand and number of the antibodies used in the Western blot analysis in this experiment have been added. Please check the uploaded revised manuscript (Line168, line 170 and line 171). In addition, I provided the original image of the western blot in the supplementary material. Please check the image entitled “β actin of Intestine”, "Acaa1a of Intestine" and “Fabp1 of Intestine” in the zip file of the uploaded supplementary material.
Point 3: There is no reference to Figure 5 in the text. In figures 5 and 6 the legends do not indicate what the letter above the histogram bars represent. In the legend of figure 5 the last statement indicates ‘Same as below’, however there is not any additional clarifying information.
Response 3: Thank you for your careful review and comments. I have added to the article a representation of the cited Figure 5. Please check the uploaded revised manuscript (Line 247). I should have explained the meaning of the lowercase letters above the columns in the legend to Figure 5 in my initial manuscript. a-d: The lowercase letters above each bar reflect the difference between the groups. If the lowercase letters above the bars are different, it means that there is a significant difference between the two groups (P < 0.05), and if there are the same letters, it means that there is no significant difference between the two groups (P > 0.05). Alternatively, the lowercase letters above the bars in Figure 6 represent the same meaning. I have completed this detailed explanation in the revised version. Please check the uploaded revised manuscript (Lines 265-269).
Point 4: Why were only Fabp1 and Acaa1a protein expression verified by western blot?
Response 4: The purpose of our Western Blot test at the end of the experiment was to verify if the results of the proteomic analysis were consistent with the results of our quantification of protein expression using antigen-antibody reactions after the proteins had been separated by SDS-PAGE. As a result, the results were convincing and we got the same results. Moreover, these two proteins were also the significantly different proteins that we filtered for in the proteomics results. However, if we were to investigate the mechanism of action of betaine in intestinal cells in a cell-based assay, we would select more relevant proteins for the Western Blot assay.
